**Subject Category:**
Biology (whole organism)

behaviour/ecology/evolution

endosymbionts, haplodiploidy, interpopulation variation, maternal effects, mating behaviour, parental effects

**Author for correspondence:**
Peter Schausberger
e-mail: peter.schausberger@univie.ac.at

# Spider mite mothers adjust reproduction and sons' alternative reproductive tactics to immigrating alien conspecifics

Peter Schausberger[1,2], Tetsuo Gotoh[3,4] and Yukie Sato[1]

[1]Sugadaira Research Station, Mountain Science Center, University of Tsukuba, Ueda, Nagano, Japan
[2]Department of Behavioural Biology, University of Vienna, Vienna, Austria
[3]Faculty of Agriculture, Ibaraki University, Ami, Ibaraki, Japan
[4]Faculty of Economics, Ryutsu Keizai University, Ryugasaki, Ibaraki, Japan

(iD) PS, 0000-0002-1529-3198

Maternal effects on environmentally induced alternative reproductive tactics (ARTs) are poorly understood but likely to be selected for if mothers can reliably predict offspring environments. We assessed maternal effects in two populations (Y and G) of herbivorous arrhenotokous spider mites *Tetranychus urticae*, where males conditionally express fighting and sneaking tactics in male–male combat and pre-copulatory guarding behaviour. We hypothesized that resident mothers should adjust their reproduction and sons' ARTs to immigrating alien conspecifics in dependence of alien conspecifics posing a fitness threat or advantage. To induce maternal effects, females were exposed to own or alien socio-environments and mated to own or alien males. Across maternal and sons' reproductive traits, the maternal socio-environment induced stronger effects than the maternal mate, and G-mothers responded more strongly to Y-influence than vice versa. G-socio-environments and Y-mates enhanced maternal egg production in both populations. Maternal exposure to G-socio-environments demoted, yet maternal Y-mates promoted, guarding occurrence and timing by sons. Sneakers guarded earlier than fighters in Y-environments, whereas the opposite happened in G-environments. The endosymbiont *Cardinium*, present in G, did not exert any classical effect but may have played a role via the shared plant. Our study highlights interpopulation variation in immediate and anticipatory maternal responses to immigrants.

# 1. Background

Variation of phenotypic traits among individuals within populations may be due to genetic differences and/or phenotypic plasticity, i.e. different trait expression by the same genotype depending on environmental factors. For phenotypic traits that are under strong sexual selection via male–male competition and/or female choice, instability and disproportionally low success of intermediate states set the stage for the evolution of diverging trait optima within the same sex. Disruptive selection leads to the expression of distinct reproductive strategies, dubbed alternative reproductive tactics (ARTs) [1–5]. ARTs may be apparent in behavioural, morphological and/or physiological traits that have been observed in numerous animal taxa and diverse mating systems, and may occur during all phases of the reproductive process. ARTs may be purely genetically determined (rare), conditionally expressed (frequent) or a combination of both (frequent) [2,3,6]. ARTs may occur in either sex but, across animals, have been far more often observed in males than females. Typically, male ARTs are dichotomous with a high-energy-investing phenotype such as territorial defender, advertiser or fighter, and a low-energy-investing phenotype parasitizing exploitation of resources, avoiding costly intrasexual interactions and/or sneaking copulations [3,6].

Our understanding of the environmental variables driving the expression of conditional male ARTs is relatively well developed for immediate (intragenerational) factors. Among other, mutually non-exclusive, immediate factors, conditional ART expression is commonly determined by ART frequency, population density, operational sex ratio and/or resource availability [3]. Transgenerational influences on ART expression, such as maternal effects, have been rarely experimentally addressed and are poorly understood [7,8]. Maternal effects are phenotypic influences of mothers on offspring phenotypes and highly relevant for both maternal and offspring fitness [9–12]. Viewing sons' ARTs and associated fitness benefit-cost trade-offs from the maternal perspective may decisively advance our understanding of the evolutionary stability and maintenance of male ARTs. Maternal effects represent ubiquitous drivers of offspring phenotypic plasticity and rank among the most important pathways shaping the phenotypic performance of offspring [11]. The adaptive significance of maternal effects depends on how good mothers are at anticipating the future environment of their offspring, which, in turn, depends on the reliability of environmental cues experienced by mothers. Therefore, maternal effects are especially likely to occur in systems with high reliability of transgenerational matching between environmental conditions experienced by mothers and their offspring, allowing mothers to adjust the offspring phenotype to match the environmental conditions offspring are likely to experience [13].

Here, we assessed maternal effects induced by interpopulation interactions on the expression of conditional ARTs in two-spotted spider mites *Tetranychus urticae* Koch. *Tetranychus urticae* is a highly polyphagous herbivore with more than 1000 reported host plant species and worldwide distribution [14]. *Tetranychus urticae* has an arrhenotokous mating system, i.e. fertilized eggs give rise to diploid daughters, whereas haploid males arise from unfertilized eggs [15]. First male sperm precedence selected for male pre-copulatory guarding behaviour and male–male fighting [16–18]. Male ARTs in *T. urticae* include a high-energy-investing fighting and a low-energy-investing sneaking phenotype [19,20]. Fighters and sneakers differ in behaviour/physiology, but not body size and morphology, and are conditionally reversible phenotypes [19]. Both male phenotypes actively search for potential mates and are mainly attracted by pheromones emitted by premature females in their final quiescent phase, called teleiochrysalis (hereafter called T-females) [21,22]. When located, both male phenotypes guard T-females by sitting on top of them, or on the females' side, touching them with their first pair of legs. Fighter phenotypes fight with other males to gain access to, and monopolize, T-females and challenge guarding fighter males to obtain the guarding position. Sneaker phenotypes do not fight with other males and are cryptic to, and thus not fought by, other males [19,20,23]. Guarding is highly important for fertilization success since copulation takes place immediately upon the emergence of mature females. The success of sneakers depends on speed in securing the guarding position and crypsis to fighters; the success of fighters mainly depends on strength in fighting and defending the guarding position.

Interpopulation interactions are common in *T. urticae*. High genetic variability and adaptability favour divergence of (sub-)populations and host race formation [24–26], which may, or may not, be accompanied by reproductive incompatibilities [27,28]. (Sub-)populations of *T. urticae* may geno- and phenotypically differ at small spatial scales such as on different neighbouring plant species [28,29]. Overlapping habitats and wind-borne medium- to long-distance dispersal [30] promote interpopulation interactions and mutual invasions. The rationale of our study was based on the

scenario that individuals of two populations, one resident, one alien, come in physical contact on a shared host plant. *Tetranychus urticae* has been shown to possess kin recognition abilities allowing them to discriminate individuals from their own and alien populations [31,32]. Resident females may perceive the arrival of alien conspecifics as threat or advantage, depending on a number of factors. These include genetic (in)compatibility, and the directionality of (in)compatibility if uni-directional, competitive ability, and expected direct and indirect fitness benefits and costs. Direct fitness effects may, for example, be mediated by changes in host plant exploitability and chemistry [33–35] or via mating in reproduction (for example more/less offspring [36]) or via indirect genetic effects [37,38]. Indirect fitness effects may comprise genetic benefits/costs via, for example, increased/decreased offspring viability or altered performance and attractiveness as mates [36]. We hypothesized that females perceiving the arrival of alien conspecifics should adjust their reproduction, transfer information about alien conspecifics to the next generation and adjust the ARTs of sons. Maternal adjustment of sons' ARTs may be apparent in altered guarding eagerness, ART ratio and/or timing of guarding, all of which are affecting the relative reproductive success of ARTs [39]. If immigrating alien conspecifics increase the fitness of residents via genetic and/or environmental effects, we predicted unchanged or worsened within-resident performance by ARTs. By contrast, if immigrating alien conspecifics are detrimental and threaten to decrease the fitness of residents via genetic, phenotypic and/or environmental effects, we predicted optimized within-resident performance by ARTs [39], to prevent or mitigate loss of their genetic, phenotypic and/or environmental advantage in the absence of alien conspecifics.

We tested our hypotheses and predictions by mutually exposing females of two allopatric, and thus mutually alien, populations of two-spotted spider mites, *T. urticae*, to own and alien socio-environments and/or own and alien mates, and quantified maternal reproduction and maternal effects on the ARTs of sons. Since reproductive interactions among arthropods, including tetranychid spider mites, may be influenced by facultative endosymbiotic bacteria such as *Wolbachia*, *Spiroplasma*, *Rickettsia* and *Cardinium* [40,41], we screened both experimental populations for endosymbiont infection before the experiment. The design of our experiment was built on the assumption that (i) being surrounded by alien conspecifics during development and oviposition, and/or (ii) mating with alien conspecifics (male mates can only exert non-genetic effects on sons because of haplodiploidy) are reliable indicators for mothers of the future environment of their offspring, and are thus likely to induce maternal effects on the ARTs of sons. Assessing the maternal response to either condition in isolation, (i) or (ii) and both conditions combined, (i) and (ii), relative to their absence (i.e. being surrounded by conspecifics and mated to males from their own population), should provide indications of the proximate pathways mediating maternal effects.

# 2. Material and methods

## 2.1. Study animals, rearing and experimental units

*Tetranychus urticae* (red form) used in experiments came from two populations, named Y- and G-population, maintained in the laboratory at Sugadaira Research Station (Ueda, Nagano, Japan). The Y-population had been founded by specimens originally obtained from Koppert B.V. (Berkel en Rodenrijs, The Netherlands), 4–5 years before starting the experiment [19,20]. The G-population had been founded by specimens obtained from Tetsuo Gotoh (Ibaraki University) two months before starting the experiment. At Ibaraki University, the donor population of the G-population had been founded by specimens collected on tomato in Iida (Nagano, Japan) and maintained on detached common bean leaf discs for about 1 year prior to the experiment. At Sugadaira Research Station, the populations were separately reared on detached primary leaves of common bean, *Phaseolus vulgaris* var. Naga-Uzura. Leaves were clipped from two to three weeks-old plants, grown in a peat moss/cocoa fibre/egg shell substrate mixture inside 9 cm Ø pots, and placed adaxial side up on moist cotton pads inside styrofoam trays (12.5 × 18 cm). To maintain cotton moisture, styrofoam trays were perforated on the bottom and floated inside larger plastic trays (23 × 32 cm) half-filled with water. Every two to three weeks, the spider mites were transferred onto new leaves.

(Pre-)experimental units included styrofoam trays, squared leaf arenas and circular leaf discs. Leaf arenas consisted of 4 × 4 cm squares, cut from detached primary bean leaves and placed adaxial side up on moist cotton wool inside styrofoam trays. Circular leaf discs (1 cm Ø) were punched out from the primary bean leaves, using a cork borer, and placed adaxial side up on moist cotton pads inside

styrofoam trays or inside acrylic cages. Acrylic cages consisted of closed Petri dishes (each 5 cm Ø, height 1.5 cm) having a mesh-covered ventilation opening (1.3 cm Ø; mesh openings 0.05 mm) in the lid (SPL Life Sciences Co. Ltd, South Korea; product 310050). Care was taken to allocate leaf arenas and discs that were cut from the same leaf equally to Y- and G-treatments.

Plants were grown, and rearing, pre-experimental and experimental units were stored inside climate chambers at $25 \pm 1°C$ and $16:8$ h light : dark.

## 2.2. Experimental procedure

To assess immediate and transgenerational influences of the maternal socio-environment (own or alien population) and maternal male mate (own or alien population) on maternal reproductive traits and ARTs of sons in two populations (the Y- and G-population) of *T. urticae*, we used a full 2 (maternal origin; Y or G) × 2 (socio-environment; Y or G) × 2 (male mate; Y or G) factorial design, resulting in eight maternal treatments (figure 1).

The experiment consisted of three major phases: (i) generating experimental mothers and exposing them during juvenile development to own or alien socio-environments, (ii) mating of experimental mothers with own or alien males and subsequent production of experimental sons in own or alien socio-environments, (iii) subjecting the experimental sons to behavioural assays.

(i)   Experimental mothers from the Y- and G-populations were, from the egg stage through juvenile development, exposed to a socio-environment consisting of individuals from their own or the alien population. To generate experimental mothers from the Y- and G-populations, females (i.e. grandmothers of the experimental sons) in their final pre-adult phase (T-females), and adult males were randomly withdrawn from the rearing and placed in groups of four females (quadruplets) and two males, all from the same population, on leaf discs (experimental (e-)day 0). After one day (e-day 1), adult females had emerged and were mated. Quadruplets of mated females were transferred onto squared leaf arenas and left to oviposit for six days, after which the females were removed (e-day 7). Developing offspring, to become experimental mothers, were exposed to own or alien socio-environments, by adding 20 own or alien males onto each arena (e-day 7). Own and alien males were removed when the offspring, to become experimental mothers, had reached the teleiochrysalis stage (e-day 12).

(ii)  T-females, to become experimental mothers, were collected from leaf arenas and placed in quadruplets on leaf discs together with two own or alien males (e-day 13). After one day, adult females had emerged and were mated. Quadruplets of mated females ($N = 10$ for each treatment) were transferred onto leaf arenas harbouring approximately 20–30 own or alien larvae and protonymphs, i.e. ovipositing experimental females were again exposed to own or alien socio-environments (the same type they had been exposed to during juvenile development; e-day 14). Ovipositing experimental females and juveniles (which created own or alien socio-environments) were removed before the eggs laid by the experimental females started to hatch (e-day 19). Thus, offspring of experimental females including experimental sons experienced only similarly aged own conspecifics but not the socio-environment experienced by their mothers (including chemical cues on the web because vanishing within less than 2 h [42]). Eggs laid by experimental females over 6 days were counted and left to develop on leaf arenas until reaching the teleiochrysalis stage (e-day 24). Teleiochrysalis (T-) males, to become experimental sons, were collected and placed in quintuplets on leaf discs for one day (e-day 25). Non-collected individuals were left on leaf arenas until reaching adulthood, to determine the offspring sex ratio for each of the eight treatments (figure 1).

(iii) To assess the ARTs of experimental sons, males (less than 24 h old) from each of the eight treatments (figure 1) were placed in groups of five, together with one T-female from the same population (Y- or G-population), randomly taken from the rearing, on a fresh leaf disc inside an acrylic cage (e-days 26–28). The ratio of five males to one T-female was chosen because of opening the greatest chance to detect the rare sneaking phenotype [20]. Every 45 min, the cages were monitored for male moving activity (ambulating), grouping behaviour and the ART of the first three guarding males for 4.5 h at maximum. Males were scored grouping when sitting together (away from the T-female) at an inter-individual distance of ≤0.5 body length. Guarding males were challenged by the 'brush test' [19] to determine their ART. The 'brush test' consists of picking up a male at the tip of his idiosoma, using a moistened brush (marten's hair, size 0), and letting the lifted male challenge the guarding male with his first

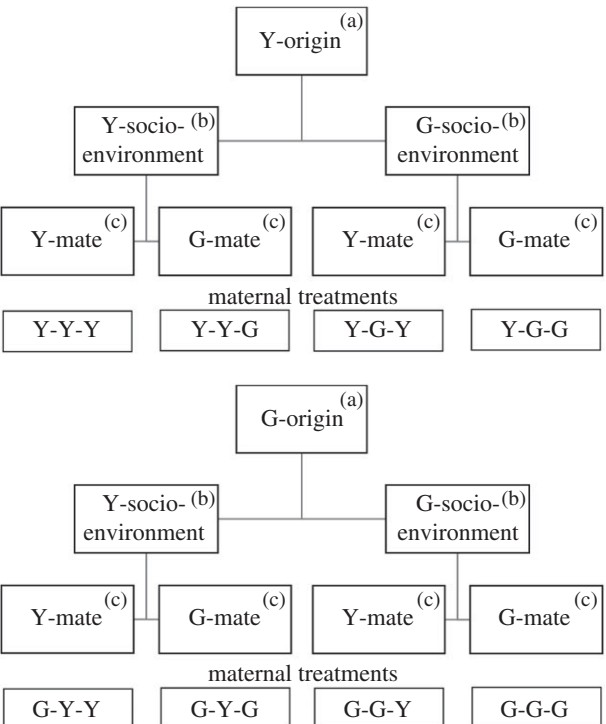

**Figure 1.** Design used to create eight maternal treatments of experimental sons. Females, to become mothers of experimental sons, (a) originated from two populations (Y or G), (b) were exposed from the egg stage through juvenile development and during oviposition to a socio-environment composed of individuals from their own or the alien population (Y or G) and (c) were mated to a male from their own or the alien population (Y or G), resulting in eight maternal treatments.

pair of legs. Guarding males assuming the fighting posture, i.e. raising their first pair of legs and threatening the challenger, are considered fighters, whereas guarding males not responding to the challenging male are considered sneakers [19]. The first and second guarding males were removed after determining their ART. Each of the eight treatments was replicated 20 to 29 times.

## 2.3. Screening for reproductive endosymbionts

Single adult females of both experimental populations, Y and G, were put into 1.5 ml Eppendorf tubes containing 30 µl of PrepMan Ultra Sample Preparation Reagent (Applied Biosystems, Foster City, CA, USA) and mashed using the tip of a micropipette. The tubes were heated to 100°C for 10 min, and the solution after heating was used as DNA template of diagnostic PCR of endosymbiont infection. Primer pairs of diagnostic PCR of bacteria were *wsp*-81F/*wsp*-691R [43] and Wol16S-F/Wol16S-R [44] for *Wolbachia*, CFB-f1/CFB-r1 [45] for *Cardinium*, Rb-F/Rb-R [46] and 17KD-F/17KD-R [47] for *Rickettsia*, and 27-F [48]/TKSSsp [49] for *Spiroplasma*. Amplification conditions were 94°C for 2 min, followed by 35 cycles at 94°C for 30 s, 52°C for 30 s, 72°C for 1 min and a final extension of 7 min at 72°C. Ten females of each population, Y and G, were examined for each primer set. Negative and positive controls were included in the reaction. PCR products were run on 1.5% agarose gel in TAE/EtBr for 25 min at 100 V, and then photographed on a UV transilluminator. As positive controls, we used DNA of *Wolbachia*-infected *Oligonychus gotohi* Ehara, *Cardinium*-infected *O. gotohi*, *Rickettsia*-infected *Liposcelis bostrychophilus* Badonnel (Psocoptera: Liposcelidae) and *Spiroplasma*-infected *Tetranychus truncatus* Ehara, which were obtained by the same procedures as mentioned above. Distilled water was used as a negative control.

## 2.4. Statistical analyses

We used IBM SPSS 25 for statistical analyses (IBM, Armonk, NY, USA). Separate generalized linear models (GLMs) were used to analyse the influence of maternal origin (Y- or G-population), maternal socio-environment (Y or G) and maternal mate (Y or G) on the number of offspring produced over 6 days (normal distribution, identity link), the offspring sex ratio (binomial distribution, logit link), guarding by sons (number of guarding males out of 3 at maximum; binomial distribution, log link; counts of events),

ambulating by sons (number of males moving out of all males present on the disc; binomial distribution, log link; counts of events) and grouping by sons (number of males grouping out of all males present on the disc; binomial distribution, log link; counts of events). Before analysis, the numbers of males observed guarding, ambulating and grouping were aggregated across observations resulting in one value per replicate. The relation between the number of guards and ambulating and grouping activity was assessed within each maternal origin by linear regression. The influence of maternal origin (Y or G), maternal socio-environment (Y or G), maternal mate (Y or G) and filial ART (sneaker and fighter) on the timing of guarding was analysed by GLM (1/x; Gamma distribution, log link). ART ratios (frequencies of fighters and sneakers) within each maternal treatment were compared with the expected ratio, calculated by lumping all treatments, using $\chi^2$ tests. Occurrence and timing of the first guarding event by sons were compared among treatments within each maternal origin (Y- and G-population) by separate Cox regressions.

# 3. Results

## 3.1. Reproductive endosymbionts

Screening for infection by the endosymbiotic bacteria *Wolbachia*, *Rickettsia*, *Spiroplasma* and *Cardinium* revealed that the G-population was infected by *Cardinium*, whereas the Y-population was free of endosymbiont infection.

## 3.2. Maternal reproductive traits

G-females produced more offspring than Y-females (figure 2 and table 1). G-socio-environments were more favourable for offspring production to both G- and Y-females than were Y-socio-environments. By contrast, G-mates decreased the egg production of both G- and Y-females, as compared to Y-mates (figure 2 and table 1). Offspring sex ratio of Y-females was more biased towards daughters in alien (G-)environments, which was not the case in G-females (indicated by the significant interaction between maternal origin and socio-environment). G-females mated to alien (Y-)males produced more daughter-biased offspring than G-females mated to own (G-)males, whereas male mate origin did not influence the offspring sex ratio of Y-females (indicated by the significant interaction between maternal origin and mate) (figure 2 and table 1).

## 3.3. Reproductive behaviour by sons

Sons of G-females guarded less than sons of Y-females (figure 3*a* and table 2). G-environments demoted the guarding behaviour of sons, whereas maternal Y-mates promoted the guarding behaviour of sons (figure 3*a* and table 2). The significant interaction between maternal origin and socio-environment indicates that the demoting effect of G-environments was more pronounced in sons from G- than Y-females (figure 3*a* and table 2). Sons of Y-females ambulated more than sons of G-females (figure 3*b* and table 2). The significant two-way interactions between maternal origin and environment and maternal origin and mate indicate that sons of G-females ambulated more in alien (Y-) than own (G-)environments and when their mothers were mated to alien (Y-) than (G-)males. By contrast, environment and mate did not affect ambulation by sons of Y-females (figure 3*b* and table 2). Sons of G-females were more prone to group together than sons of Y-females (figure 3*c* and table 2). Sons of both Y- and G-females grouped less in alien than own environments and when mated to alien versus own mates, as indicated by the significant two-way interactions between origin and environment and environment and mate (figure 3*c* and table 2). In both populations (Y and G) the number of guarding males was positively correlated with the proportion of ambulating males but was unrelated to the proportion of grouping males (figure 4).

The ratio between the ARTs of sons (fighter and sneaker) only deviated from the overall mean ratio in sons from Y-females that had been exposed to a Y-environment and were mated to a male from the G-population (Y-Y-G; figure 5). Such sons were more likely to adopt the sneaking tactic than sons in all other maternal treatments. The occurrence and timing of the first guarding event did not differ among maternal treatments in sons from Y-females (figure 6). By contrast, sons from G-females were more likely to guard, and guarded earlier, when their mothers had been exposed to an alien (Y-)environment and/or were mated to an alien (Y-)mate than when their mothers had experienced a G-environment and were mated to a G-male (figure 6). Relative guarding times were only influenced

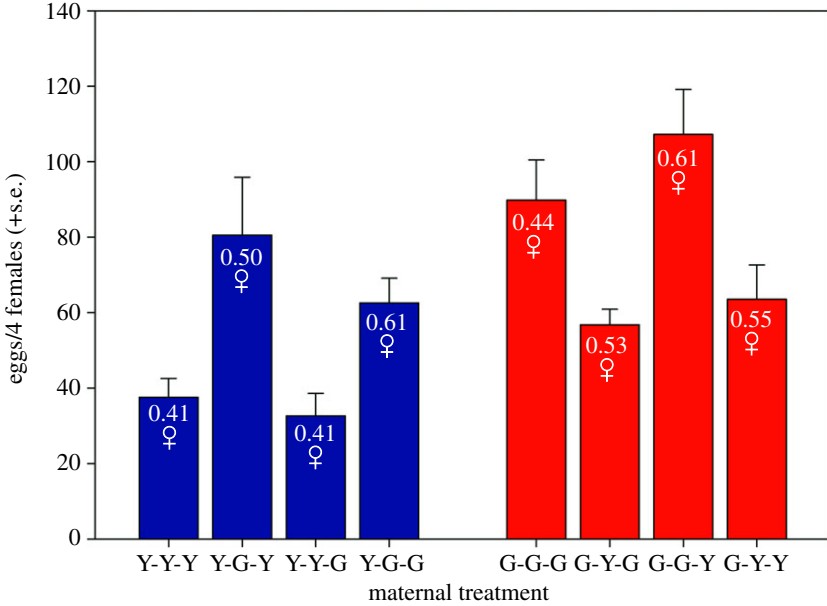

**Figure 2.** Maternal reproductive traits. The mean number of offspring and their sex ratio, produced by mothers from the Y- and G-populations, which were exposed during juvenile development and oviposition to a socio-environment composed of individuals from their own or the alien population (Y or G) and were mated to a male from their own or the alien population (Y or G). For oviposition, mothers were placed in quadruplets on leaf squares for 6 days. $N = 10$ quadruplets of females for each treatment for the number of eggs produced and 340–850 eggs per treatment for offspring sex determination. First, second and third letters of treatment acronyms refer to maternal origin, socio-environment and mate. Numbers inside bars represent the daughter proportion among offspring.

**Table 1.** Results of GLMs on the influence of population origin, socio-environment and mate on the number of offspring (normal distribution, identity link) and sex ratio (binomial distribution, log link) of experimental females. Females originated from two populations (Y or G) and were exposed during juvenile development and oviposition to a socio-environment composed of individuals from their own or the alien population (Y or G) and were mated to a male from their own or the alien population (Y or G) (figure 1).

| independent variables | offspring number | | offspring sex ratio | |
|---|---|---|---|---|
| | wald $\chi_1^2$ | $p$-value | wald $\chi_1^2$ | $p$-value |
| origin | 18.180 | <0.001 | 10.917 | 0.001 |
| environment | 37.632 | <0.001 | 16.761 | <0.001 |
| mate | 3.736 | 0.05 | 0.417 | 0.52 |
| origin × environment | 0.024 | 0.88 | 23.936 | <0.001 |
| origin × mate | 0.003 | 0.96 | 33.700 | <0.001 |
| environment × mate | 0.949 | 0.33 | 2.360 | 0.12 |

by the interaction between the environment and ART (fighter and sneaker) (GLM; Wald $\chi_1^2 = 6.744$, $p = 0.009$; for all other main effects and two-way interactions Wald $\chi_1^2 \leq 2.694$, $p > 0.10$). In Y-environments, sneakers guarded earlier than fighters whereas the opposite, fighters guarding earlier than sneakers, was the case in G-environments (figure 7).

## 4. Discussion

Our study confirms the predictions that mothers should respond to the immigration of alien conspecifics by adaptively adjusting their reproduction and, via maternal effects, the ARTs of sons to the anticipated

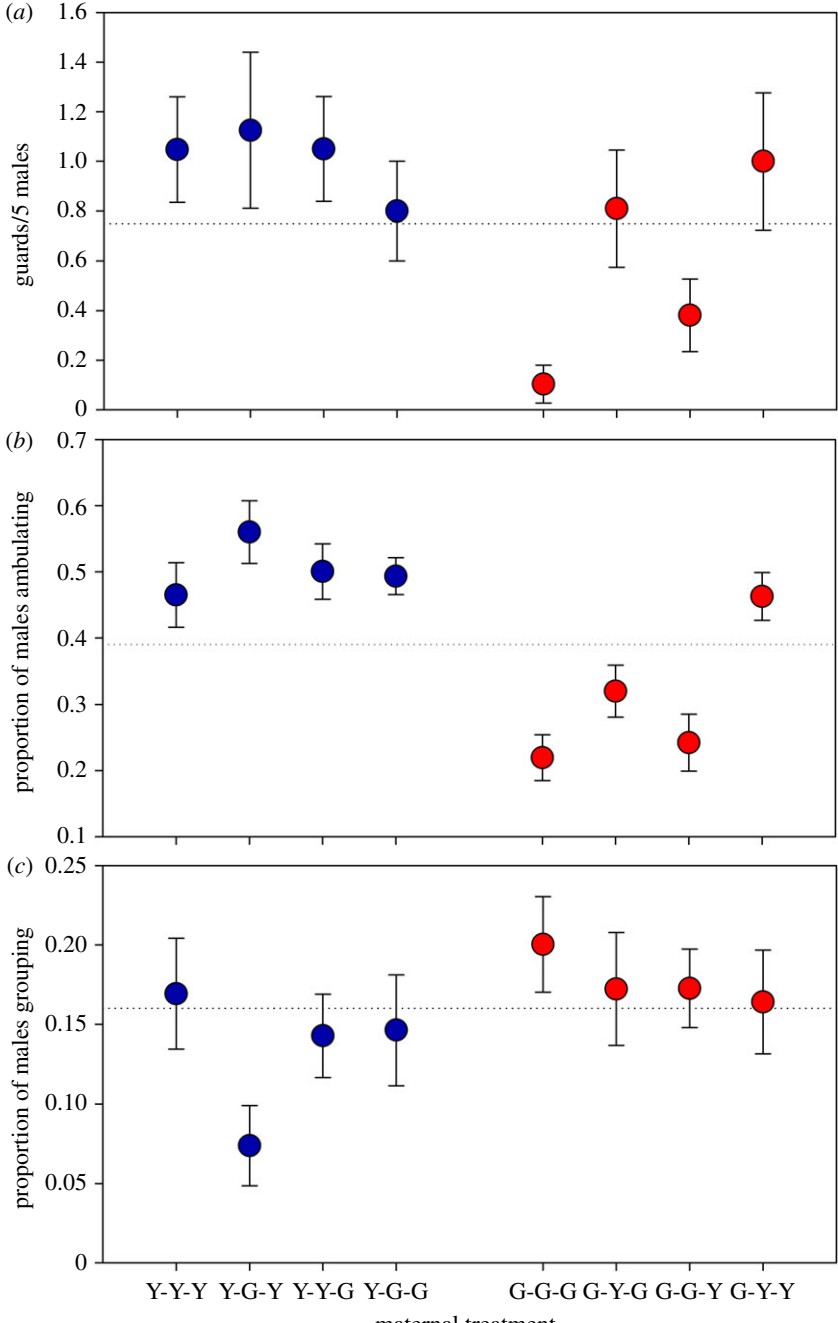

**Figure 3.** Filial behavioural traits. Guarding (*a*), ambulating (*b*) and grouping (*c*) by sons emerging from mothers from the Y- or G-population, which were exposed during juvenile development and oviposition to a socio-environment composed of individuals from their own or the alien population (Y or G) and were mated to a male from their own or the alien population (Y or G). First, second and third letters of treatment acronyms refer to maternal origin, socio-environment and mate. Sons were held in groups of five together with a teleiochrysalis female, all from the same population, on a leaf disc and their behaviour was observed every 45 min until three guarding events occurred for 270 min at maximum ($N = 20–29$ for each treatment). Dotted horizontal lines indicate the overall means calculated by lumping all treatments.

future environment [13]. Both (i) baseline reproduction and sons' behaviour (within the same population), and (ii) interpopulation interaction-induced alterations of maternal reproduction and maternal effects on ARTs of sons were largely population-specific and depended on whether exposure to alien conspecifics happened via socio-environmental factors and/or male mates. Across traits, the socio-environment (own or alien population) exerted stronger effects than male mate origin (from own or alien population); Y- and G-socio-environments and Y- and G-mate origins acted in opposite directions for maternal reproduction, in the same direction for the guarding behaviour by sons and

**Table 2.** Results of GLMs (binomial distribution, log link; counts of events) on guarding, ambulating and grouping by experimental sons, as influenced by maternal origin, socio-environment and mate. Mothers of experimental sons came from two populations (Y or G) and were exposed during juvenile development and oviposition to a socio-environment composed of individuals from their own or the alien population (Y or G) and were mated to a male from their own or the alien population (Y or G) (figure 1).

| independent variables | guarding | | ambulating | | grouping | |
|---|---|---|---|---|---|---|
| | wald $\chi_1^2$ | $p$-value | wald $\chi_1^2$ | $p$-value | wald $\chi_1^2$ | $p$-value |
| origin | 18.647 | <0.001 | 171.342 | <0.001 | 13.545 | <0.001 |
| environment | 15.729 | <0.001 | 21.692 | <0.001 | 2.235 | 0.13 |
| mate | 4.882 | 0.02 | 8.832 | 0.003 | 1.334 | 0.25 |
| origin × environment | 11.364 | 0.001 | 49.007 | <0.001 | 7.509 | 0.006 |
| origin × mate | 1.227 | 0.27 | 9.474 | 0.002 | 0.238 | 0.63 |
| environment × mate | 2.419 | 0.12 | 0.137 | 0.71 | 9.428 | 0.002 |

interacted for other traits. Interactions between maternal origin and male mate were observed for offspring sex ratio and ambulating behaviour of sons. We argue that the nature of the resident maternal response in own reproduction and information transmission to sons (maternal effects) depended on whether the females perceived alien conspecifics as a threat, decreasing resident maternal and sons' fitness, or advantage, increasing their fitness.

Judgement of whether the immigrating alien conspecifics posed a threat or advantage to residents, and interpreting the contrasting population-specific responses to aliens, comes from comparing the baseline characteristics of each population, i.e. when all maternal treatment factors are aligned (Y-Y-Y and G-G-G). In relative comparison (Y-Y-Y versus G-G-G), the G-population was, without extrinsic influence by alien conspecifics, characterized by higher reproductive output, and extremely low guarding and ambulating, yet high grouping, propensity by sons. Females of both populations produced twice as many eggs in socio-environments of G-individuals than in socio-environments of Y-individuals. Thus, resident Y-females benefited from a favourable socio-environment created by G-individuals, whereas resident G-females faced a detrimental socio-environment created by Y-individuals. Conversely, males from the Y-population were superior mates to males from the G-population, enhancing the number of eggs produced by G- and Y-females in either socio-environment.

## 4.1. Maternal reproductive traits: a role for *Cardinium*?

*Cardinium* are widespread endosymbiotic bacteria of arthropods, including spider mites [41], with often severe effects on the reproduction of their hosts such as causing cytoplasmic incompatibility (CI), feminization or parthenogenesis [50]. No such effects were apparent in the *Cardinium*-infected G-population or crossings between G and Y individuals. However, many other miscellaneous effects of facultative endosymbionts on the physiology and behaviour of their hosts are possible [50]. Uninfected Y-males were superior mates to both uninfected Y- and *Cardinium*-infected G-females, but there were no apparent incompatibilities between mates of the two populations, no distorted sex ratio or decreased egg hatchability, that could be assigned to *Cardinium* infection in males of the G-population. Rather than mediated by *Cardinium*, the positive effect of Y-males on the reproduction of females from both populations was caused by interpopulation differences in in- and out-breeding depression/advantages arising from differences in intrapopulation inbreeding levels [51–53]. Judged from the number of eggs produced by mating with own or alien males, Y-females had an inbreeding, yet G-females an outbreeding, advantage. Y-males being superior mates for females of both populations and the assumption of optimal in- and out-breeding balance [51] suggest a lower inbreeding level and higher genetic variability within the Y- than G-population.

In contrast with the influence of male mates, the positive effect of G-socio-environments on females of both the Y- and G-populations may well have been mediated by *Cardinium*, via favourably altering leaf chemistry or by transmission from G- to Y-individuals via feeding on the same leaf. It is known that endosymbionts of herbivores, such as *Wolbachia* and *Cardinium*, may

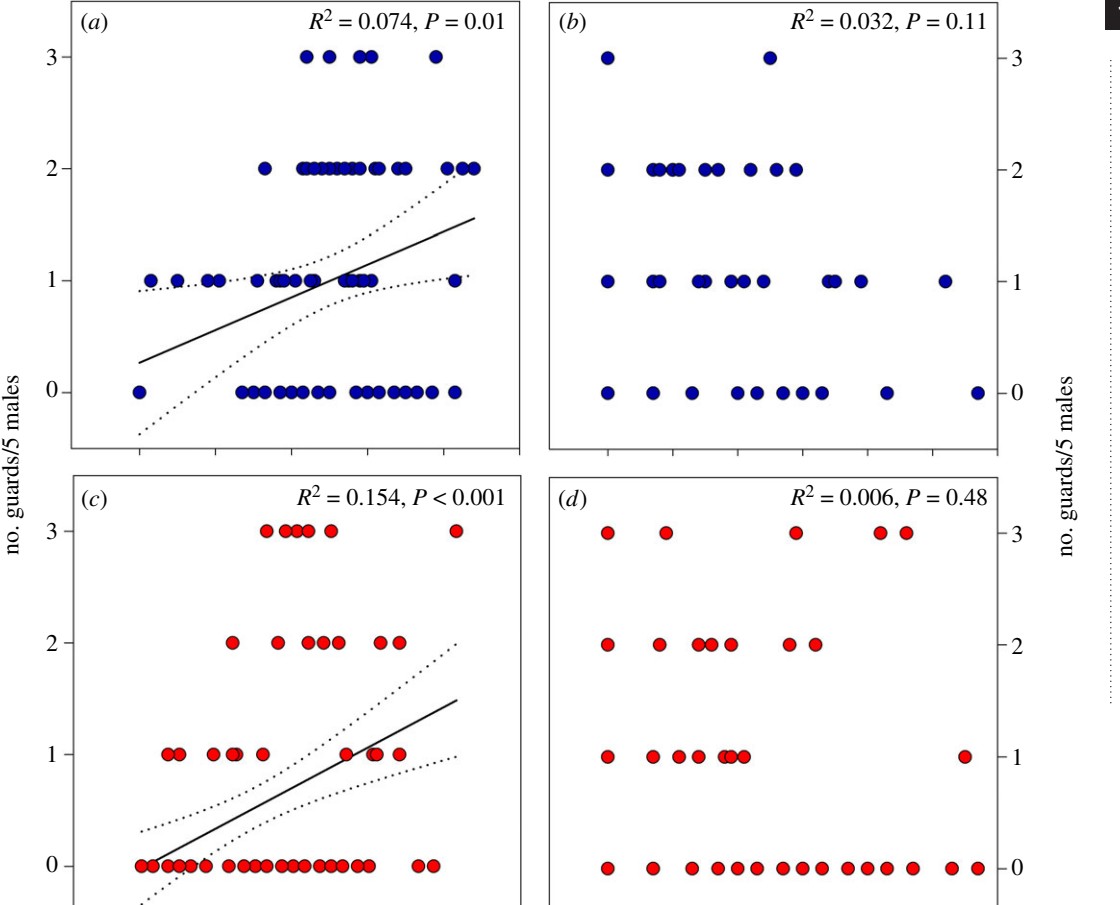

**Figure 4.** Relation between guarding and ambulating/grouping. Number of guarding sons linearly regressed (±95% CI) on proportions of males ambulating (*a,c*) and grouping (*b,d*). Sons were produced by mothers originating from the Y- (*a,b*) or G-population (*c,d*). Mothers were exposed during juvenile development and oviposition to a socio-environment composed of individuals from their own or the alien population (Y or G) and were mated to a male from their own or the alien population (Y or G). Sons were held in groups of five together with a teleiochrysalis female, all from the same population, on a leaf disc and their guarding, ambulating and grouping behaviour observed every 45 min until three guarding events occurred for 270 min at maximum (*N* = 20–29 for each treatment).

change the leaf chemistry to the benefit of their herbivorous hosts by suppressing induced direct anti-herbivore plant defence [55] or retarding leaf ageing by modifying the phytohormonal status such as increasing cytokinin levels [56]. Endosymbionts may also be transmitted between infected and uninfected herbivorous individuals via the shared host plant (*Cardinium* [57] and *Wolbachia* [58]). Considering that *P. vulgaris* is poorly directly defended against herbivores (argument against putative suppression of inducible direct anti-herbivore defence), that G-individuals did worse in Y-environments (argument against inter-individual transfer of *Cardinium* via the leaf), we argue that feeding by *Cardinium*-infected G-individuals rendered the bean leaves nutritionally more favourable than feeding by endosymbiont-free Y-individuals. Less likely, but not completely excludable, proximate mechanisms are direct favourable environmental effects from the G- on the Y-population. Chemosensory and/or visual cues on G-individuals or their webbing may have exerted positive effects on feeding behaviour and resulting oviposition by females of both populations. The webbing by spider mites is considered an important source of information and spider mites are known to possess sophisticated abilities to discriminate conspecific webs based on population origin [31], life stage [32] or familiarity [53]. Non-leaf-mediated effects of genetic relatedness and density (via physical contact between individuals or the web) on the dispersal behaviour of *T. urticae* were shown by Bitume *et al.* [59].

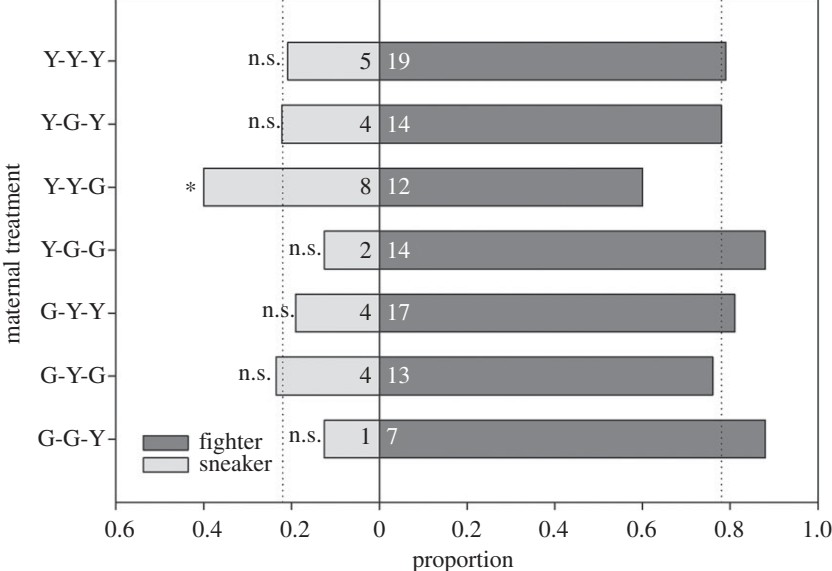

**Figure 5.** Filial ARTs. Ratio of fighter and sneaker phenotypes among sons produced by mothers from the Y- or G-population. Mothers were exposed during juvenile development and oviposition to a socio-environment composed of individuals from their own or the alien population (Y or G) and were mated to a male from their own or the alien population (Y or G). Sons were held in groups of five together with a teleiochrysalis female, all from the same population, on a leaf disc and their guarding behaviour observed every 45 min until three guarding events occurred for 270 min at maximum ($N = 20$–29 for each treatment). The first and second guards were removed after determining their ART. The first, second and third letters of treatment acronyms refer to maternal origin, socio-environment and mate, respectively. Dotted vertical lines represent the expected mean ratio calculated by lumping all treatments. Within each maternal treatment, fighter and sneaker frequencies were compared with the expected frequencies by chi-square tests (* for $p < 0.05$, ns for non-significant). G-G-G sons were excluded from the analysis because only three guards (all fighters) occurred.

## 4.2. Reproductive behaviour by sons

Sons of G-mothers, who had experienced a Y-environment and/or were mated to a Y-male, were more eager to guard own (G-)females than sons of baseline (G-G-G) mothers. This maternally induced behavioural alteration in haploid sons is adaptive because it promotes the persistence of the G-phenotype, rendering the environment more favourable for offspring. Whether or not this maternal effect on sons' behaviour was proximately driven by *Cardinium* influencing maternal hormonal status and provisioning remains to be tested but is well conceivable because such a behavioural change benefits *Cardinium*, warranting its persistence and promoting its spread. Endosymbionts may alter male competitive behaviour and mating rates to their benefit, as has been shown, for example, for *Wolbachia* infection in *Drosophila* [60,61]. By contrast, Y-mothers had no reason to modify the (already high) guarding eagerness of sons when experiencing alien (G-)socio-environments or alien mates because alien (G-)socio-environments were more favourable than their own (Y-)socio-environments. Also, the timing of guarding by sneakers and fighters fits to this explanation. In Y-socio-environments (considered benign regarding threat posed by immigrants), sneakers guarded earlier than fighters (as did sons of unmated and sneaker-mated females), whereas in G-socio-environments (excluding the baseline G-G-G treatment because in total only three fighters were observed), fighters guarded earlier than sneakers (as did sons of fighter-mated mothers). Heavy increase in guarding eagerness of own (G-)females can be seen as a counter-response to the unfavourable Y-environment putting offspring performance at stake. Retarding the timing of guarding of sneakers relative to that of fighters can be seen as reduced maternal investment in sneakers and increased investment in fighters in anticipation of out-competing unfavourable alien conspecifics.

Positive correlations between ambulating behaviour and guarding propensity in both populations, viewed in concert with lacking difference among maternal treatments in guarding propensity by sons from Y-mothers, but dramatic differences among maternal treatments in guarding propensity by sons from G-mothers point at either (i) guarding propensity being proximately mediated by stronger arrestment at T-females [22] rather than enhanced male mate searching activity, or (ii) enhanced mate searching activity resulting in higher guarding propensity in sons from G- but not Y-females. Male

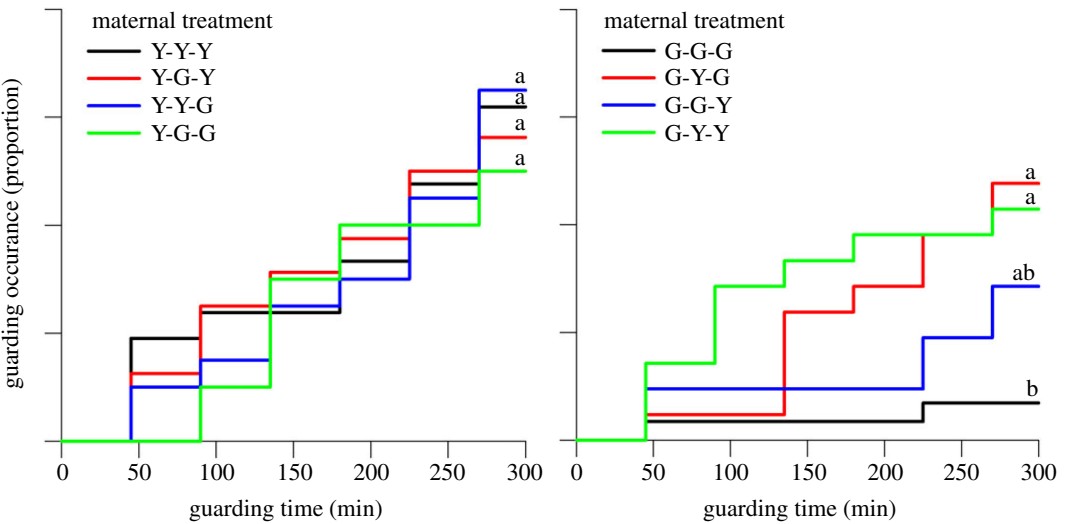

**Figure 6.** Occurrence and timing of the first guarding event by Y- and G-sons. Sons were held in groups of five together with a teleiochrysalis female, all from the same population, on a leaf disc and monitored in 45 min intervals for occurrence of the first guarding event ($N = 20$–29 for each treatment). The first, second and third letters of treatment acronyms refer to maternal origin, socio-environment and mate, respectively. Mothers of experimental sons originating from the Y- or G-population were exposed during juvenile development and oviposition to a socio-environment composed of individuals from their own or the alien population (Y or G) and were mated to a male from either their own or the alien population (Y or G). Cox regression was performed separately for sons from Y- and G-mothers: different lower-case letters indicate significant differences among maternal treatments ($p < 0.05$) within the Y- and G-populations.

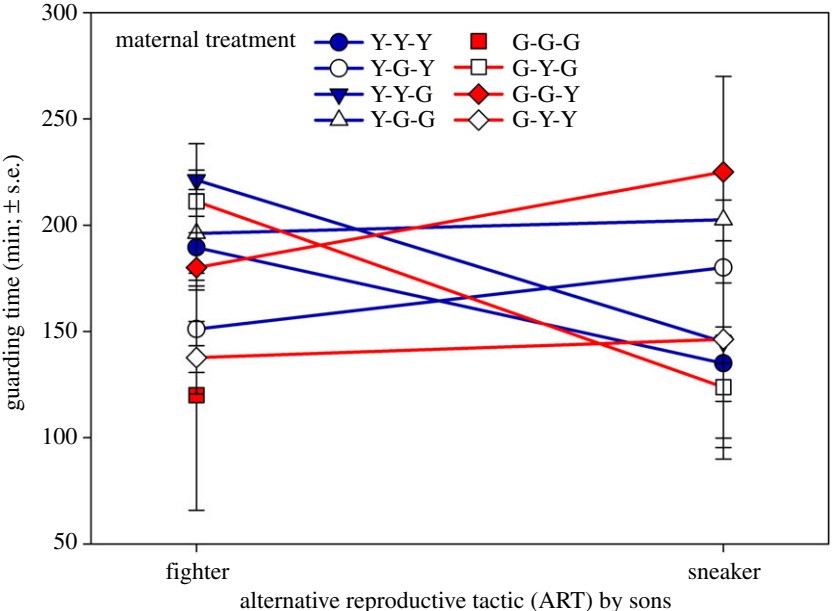

**Figure 7.** Timing of guarding by fighter and sneaker sons. Sons were held in groups of five together with a teleiochrysalis female, all from the same population, on a leaf disc and their guarding behaviour observed every 45 min until three guarding events occurred for 270 min at maximum ($N = 20$–29 for each treatment). The first and second guards were removed after determining their ART. The first and second letters of treatment acronyms refer to maternal origin, socio-environment and mate, respectively. Mothers of experimental sons, originated from the Y- or G-population, were exposed during juvenile development and oviposition to a socio-environment composed of individuals from their own or the alien population (Y or G) and were mated to a male from their own or the alien population (Y or G).

grouping tendency was overall stronger in G- than Y-males, more labile to extrinsic influence by alien conspecifics in sons from Y- than G-females, and within each population the strongest in the baseline maternal treatment (Y-Y-Y and G-G-G). Stronger grouping tendency in G- than Y-males might be

associated with *Cardinium* infection. Vala *et al.* [62] observed a *Wolbachia*-mediated effect on aggregation behaviour by *T. urticae*. Lacking correlation between male grouping and guarding propensity in both populations suggests that male grouping behaviour was driven by forces other than sexual selection, such as positive reinforcement and benefits arising from joint web production and host plant exploitation [63], and/or, if mating-related, to check out each other's strengths and mating intentions.

The ART ratio, i.e. ratio between fighters and sneakers, deviated from the overall mean ratio, by a higher sneaker proportion, only in sons from Y-mothers experiencing an own (Y-)socio-environment and mated to a G-male (treatment Y-Y-G). We assume that mothers disproportionally investing in sneakers prepare their sons for benign environments (Y-Y-G mothers produced the lowest offspring density among all maternal treatments, indicating a non-competitive offspring environment). Sneakers have a survival advantage under mild but not strong male–male competition [23]. Alternatively, this pattern could point at G-mates manipulating Y-females to produce less competitive sons to gain a selective advantage themselves via increased genetic contribution in daughters in F1 and grandsons in F2.

# 5. Conclusion

Our study addressed a previously largely neglected issue in animals showing ARTs, that is maternal effects on ARTs by sons. We show that maternal perception of alien conspecifics via the socio-environment and/or male mates makes mothers not only adjust their own reproductive behaviour but also adjust the guarding behaviour and associated behavioural traits of their sons. Proximately, the observed interpopulation differences in mutual response, maternal reproduction and maternal effects on sons may have been (co-)mediated by *Cardinium*, which was present in one but not the other population. We argue that interpopulation variation in the adjustment of own reproductive behaviour and maternal effects were ultimately driven by whether immigrating alien conspecifics posed a current and future threat or advantage to resident mothers, suggesting adaptive maternal effects.

Ethics. No ethical approval or specific permit was needed for rearing and experimental use of *T. urticae*, which is neither a protected nor endangered species.
Data accessibility. All supporting data are available in the electronic supplementary material.
Authors' contributions. P.S. conceived the study idea and designed the experiment; P.S. and Y.S. acquired funding and conducted the experiment; T.G. conducted the endosymbiont screening; P.S. analysed the data and wrote the manuscript; P.S., T.G. and Y.S. contributed to revision of the manuscript. All authors approved submission of the manuscript.
Competing interests. The authors declare to have no competing interests.
Funding. The authors are grateful to the Japan Society for the Promotion of Science (JSPS) for financially supporting this study (invitation fellowship L18534 to P.S. and KAKENHI grant no. 17K07556 to Y.S.). Open access funding was provided by the University of Vienna.

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
