## [Reviewer comments · Royal Society Open Science]

Review History

RSOS-191201.R0 (Original submission)

Review form: Reviewer 1

Is the manuscript scientifically sound in its present form?

Yes

Are the interpretations and conclusions justified by the results?

Yes

Is the language acceptable?

Yes

Do you have any ethical concerns with this paper?

No

Have you any concerns about statistical analyses in this paper?

No

Recommendation?

Accept with minor revision (please list in comments)

Comments to the Author(s)

I read with fascination the manuscript entitled Spider mite mothers adjust reproduction and sons' alternative reproductive tactics to alien conspecifics. The authors investigated an important and increasingly popular topic in evolutionary and behavioral ecology: the influences of transgenerational plasticity. Particularly, the authors investigated how female and son reproductive behavior is altered by the socio-environment and mate origin that the female experiences. The authors found compelling evidence that supports the notion that females do in fact adjust their behavior and the behavior of their (haploid) sons in response to the socio-environment and its perceived risk.

I found the underlying premise for the study intriguing, the methodology and statistics sound, and the results exciting. My only complaint is that I found the introduction fairly difficult to follow. First, I note use of terminology such as "alien" and "invasive" does not seem to fit here as the study concerns conspecifics of differing populations. Second, the build-up to the hypotheses and predictions are difficult to follow. Much of the rationale for any given prediction is not explained until the end of the manuscript, in the discussion. This makes it hard to understand how the authors intend to interpret things such as female perception of risk and what expectation (if any) for how females and males should respond to this risk. I would like to see the introduction for this manuscript modified so that a priori hypotheses are made clear.

Transgenerational influences on Alternative reproductive tactics

1. Line 36: I question if it is necessary to include the discussion of the endosymbiont here given that testing for it was standard procedure and didn't appear to influence the results
2. Line 38: The term "invader" here does not seem to capture the ecological relevance of this question. First, I'm not quite sure which colony should be considered the invader in this scenario (is it the experimental mother or the mates?). Second if we are discussing inter-population mixing, then "immigration" is probably a more accurate term.
3. Line 69: erroneous comma after plasticity.
4. Also as an aside, in lines 74-76 discusses how we often expect maternal effects to occur largely when the environment of the mother matches that of the offspring. However, line 97 states that inter-population interactions in this species are common (and presumably so is exposure to cues from different populations). If this is the case, can you explain why *T. urticae* would express maternal effects if the environments may be unpredictable? One explanation is that your design is providing very predictable information by exposing females to the same environment at two phases (T-phase and when ovipositing). This link should be made clear somewhere in the MS.
5. Line 86: citation typo after "phenotypes"
6. Line 93: This bit of the sentence is confusing. Consider rewording.
7. Line 100: Are there known behavioral changes when populations interact? Please briefly mention How/if individuals can discern from same vs distinct populations (line 106)? Also due note that your use of the term "invasion" may be better described as "immigration". Unless the (sub)populations essentially function as colonies, in which case this isn't made clear. In any case, you use the term "alien" and "invader" interchangeably throughout the MS. I recommend sticking with one ("alien" is in the title and probably more accurately describes these interactions although probably not more so than "immigrant").
8. Line 103: But this isn't at all the scenario being tested. For example, part of your experimental design is more like an immigrant female ovipositing on the host plant of a different population.

9. Line 104-107: Please consider breaking this sentence up or rephrasing. It is currently a bit difficult to read. Suggestion: “Resident females may perceive the arrival of alien conspecifics as a threat or an advantage depending on a number of factors. These include...”

10. Line 112: What is meant by females “adjust[ing] their own reproductive behavior”? My understanding up to this point is that females are essentially immobile at the time of copulation. Do females have control over reproductive output? Please clarify.

11. Line 117: What is meant by ‘performance’ by ARTs? Do you mean male mating success or the overall expression of different ARTs? And what is meant by “enhanced”? Because there were no clear differences in ART ratio

12. Line 118: Thinking about these hypotheses and predictions, I find myself conflicted. Up to this point I haven’t been convinced that mothers can perceive the presence of alien vs resident conspecific (line 112). How is this information transmitted (e.g., through silk, touching with first pair of legs, etc.)? Moreover, there should be more attention dedicated towards building these predictions. My biggest question thus far is why should we expect inter-population interactions to influence maternal effects, specifically alter son ART? The authors describe the complexity of the cost/benefits of interpopulation interactions, many of which are influenced by factors not tested here. Therefore, it is not clear why females should particularly respond in one way or another to a different population, or the direction it responds. Furthermore, it is left unclear as to how adjusting son ARTs should be in response to interpopulation interactions. For example, if the presence of an alien conspecific reduces host-plant defenses, what changes are expected to occur in female reproductive behavior/son ART?

13. Line 181: This had been previously stated in line 88

14. Line 130: Seeing now that the female perception of alien vs resident environment/mate is an assumption, it would be helpful to motivate this assumption here by referencing work showing that there is some level of discrimination in this species (e.g., Clotuche et al 2012)

15. Line 205: suggest starting a separate section here for assessment of son ARTs (the paragraph is very long)

16. Line 242: For predicting the number of offspring produced, why assume a normal distribution and not Poisson or binomial for count data?

17. Line 306: It is not clear if/how these responses are adaptive. There were changes in response to mate and socio-environment, however there is no clear link as to how adaptive these alterations are.

18. Line 318-331: I believe the logic expressed in this paragraph should be incorporated into the introduction which would alleviate a lot of my concerns (comment 12). If the assertion is that maternal effects will be expressed as an adaptive response to the risk/reward of inter-population interactions, please explain how you interpret risk/reward and the responses to each. That is, although you do not have a priori predictions as for which population of females will be at risk, you do expect those females that are at risk to be the population that produce less eggs in the new socio-environment than the resident. Given that, what alterations are expected in response to that risk/reward environment? For example, is there one reproductive tactic to be expressed over another in a given environment (lines 409-417)? This could follow right after suggestion number 11 and, in my opinion, would make this manuscript very streamlined with clear underlying predictions. I think this is a very interesting study but currently it is difficult to follow some of the logic.

19. Line 329: I am curious what you would expect if males from different populations were to compete for access to females. Here you show that Y- males appear to fertilize more eggs, but would they be able to compete directly with G- males.

Review form: Reviewer 2

Is the manuscript scientifically sound in its present form?

Yes

Are the interpretations and conclusions justified by the results?

Yes

Is the language acceptable?

Yes

Do you have any ethical concerns with this paper?

No

Have you any concerns about statistical analyses in this paper?

No

Recommendation?

Major revision is needed (please make suggestions in comments)

Comments to the Author(s)

In this paper, the authors demonstrated how maternal social-environment and mate affects their own egg production and sons' pre-mating behaviour. The full 2 x 2 x 2 factorial design is good to test their hypothesis. However, I cannot understand a couple of experimental designs. Why did the authors expose mothers to their own or alien conspecifics twice (i.e. during juvenile development and during oviposition)? Why not only during oviposition? And why were mothers exposed to 20 males during juvenile development (line 187)? This is an unusual situation for this mite. I think it would be more convincing if the authors expose mothers to their own or alien conspecifics only during oviposition. The current experimental design is a bit tricky.

In addition, Figure 1 does not give proper information to readers. Can the authors show the treatments with the time course? Also, can they show maternal origin and treatments separately? Exactly saying, the first letter, G and Y, of Y-Y-Y, Y-G-Y, Y-Y-G...etc. are not 'maternal treatments', but 'maternal origin'. Y-Y-Y, Y-G-Y, Y-Y-G...etc. look like there were three treatments.

The colours in Figures are also confusing. In Figures 2-4 & 7, blue colour is used for Y maternal origin and red colour is for G maternal origin, but it is not clear in Figure 1. And if the authors use blue and red colours for G and Y, it is better to use different colours for Figure 5.

Decision letter (RSOS-191201.R0)

28-Aug-2019

Dear Dr Schausberger,

The editors assigned to your paper ("Spider mite mothers adjust reproduction and sons' alternative reproductive tactics to alien conspecifics") have now received comments from reviewers. We would like you to revise your paper in accordance with the referee and Associate Editor suggestions which can be found below (not including confidential reports to the Editor). Please note this decision does not guarantee eventual acceptance.

Please submit a copy of your revised paper before 20-Sep-2019. Please note that the revision deadline will expire at 00.00am on this date. If we do not hear from you within this time then it will be assumed that the paper has been withdrawn. In exceptional circumstances, extensions may be possible if agreed with the Editorial Office in advance. We do not allow multiple rounds of revision so we urge you to make every effort to fully address all of the comments at this stage. If deemed necessary by the Editors, your manuscript will be sent back to one or more of the original reviewers for assessment. If the original reviewers are not available, we may invite new reviewers.

- Data accessibility

<http://datadryad.org/submit?journalID=RSOS&manu=RSOS-191201>

- Competing interests

- Authors' contributions

All submissions, other than those with a single author, must include an Authors' Contributions section which individually lists the specific contribution of each author. The list of Authors

should meet all of the following criteria; 1) substantial contributions to conception and design, or acquisition of data, or analysis and interpretation of data; 2) drafting the article or revising it critically for important intellectual content; and 3) final approval of the version to be published.

- Acknowledgements

- Funding statement

on behalf of Dr Alexander Ophir (Associate Editor) and Kevin Padian (Subject Editor)
openscience@royalsociety.org

Associate Editor's comments (Dr Alexander Ophir):

Dear Dr. Schausberger,

I have received the comments from two expert reviewers in your field, whose comments are appended below. Each has raised some points that you will need to address before this paper can move forward. Reviewer 1 has provided several minor comments and, in particular, would like to see the writing and clarity in the introduction improved. Reviewer 2 has provided you with fewer comments, but indicated some important hesitations about your design. You will either need to conduct the additional experiment they suggest or sufficiently convince them that this experiment is unnecessary by addressing this question. I hope you are able and willing to revise your manuscript accordingly.

Best
Alex Ophir
Associate Editor, RSOS

Comments to Author:

Reviewers' Comments to Author:

Reviewer: 1

Comments to the Author(s)

I read with fascination the manuscript entitled Spider mite mothers adjust reproduction and sons' alternative reproductive tactics to alien conspecifics. The authors investigated an important and increasingly popular topic in evolutionary and behavioral ecology: the influences of transgenerational plasticity. Particularly, the authors investigated how female and son reproductive behavior is altered by the socio-environment and mate origin that the female experiences. The authors found compelling evidence that supports the notion that females do in fact adjust their behavior and the behavior of their (haploid) sons in response to the socio-environment and its perceived risk.

I found the underlying premise for the study intriguing, the methodology and statistics sound, and the results exciting. My only complaint is that I found the introduction fairly difficult to follow. First, I note use of terminology such as "alien" and "invasive" does not seem to fit here as the study concerns conspecifics of differing populations. Second, the build-up to the hypotheses and predictions are difficult to follow. Much of the rationale for any given prediction is not explained until the end of the manuscript, in the discussion. This makes it hard to understand how the authors intend to interpret things such as female perception of risk and what expectation (if any) for how females and males should respond to this risk. I would like to see the introduction for this manuscript modified so that a priori hypotheses are made clear.

Transgenerational influences on Alternative reproductive tactics

1. Line 36: I question if it is necessary to include the discussion of the endosymbiont here given that testing for it was standard procedure and didn't appear to influence the results
2. Line 38: The term "invader" here does not seem to capture the ecological relevance of this question. First, I'm not quite sure which colony should be considered the invader in this scenario (is it the experimental mother or the mates?). Second if we are discussing inter-population mixing, then "immigration" is probably a more accurate term.
3. Line 69: erroneous comma after plasticity.
4. Also as an aside, in lines 74-76 discusses how we often expect maternal effects to occur largely when the environment of the mother matches that of the offspring. However, line 97 states that inter-population interactions in this species are common (and presumably so is exposure to cues from different populations). If this is the case, can you explain why *T. urticae* would express maternal effects if the environments may be unpredictable? One explanation is that your design is providing very predictable information by exposing females to the same environment at two phases (T-phase and when ovipositing). This link should be made clear somewhere in the MS.
5. Line 86: citation typo after "phenotypes"
6. Line 93: This bit of the sentence is confusing. Consider rewording.
7. Line 100: Are there known behavioral changes when populations interact? Please briefly mention How/if individuals can discern from same vs distinct populations (line 106)? Also due note that your use of the term "invasion" may be better described as "immigration". Unless the (sub)populations essentially function as colonies, in which case this isn't made clear. In any case, you use the term "alien" and "invader" interchangeably throughout the MS. I recommend sticking with one ("alien" is in the title and probably more accurately describes these interactions although probably not more so than "immigrant").
8. Line 103: But this isn't at all the scenario being tested. For example, part of your experimental design is more like an immigrant female ovipositing on the host plant of a different population.
9. Line 104-107: Please consider breaking this sentence up or rephrasing. It is currently a bit difficult to read. Suggestion: "Resident females may perceive the arrival of alien conspecifics as a threat or an advantage depending on a number of factors. These include..."

10. Line 112: What is meant by females “adjust[ing] their own reproductive behavior”? My understanding up to this point is that females are essentially immobile at the time of copulation. Do females have control over reproductive output? Please clarify.
11. Line 117: What is meant by ‘performance’ by ARTs? Do you mean male mating success or the overall expression of different ARTs? And what is meant by “enhanced”? Because there were no clear differences in ART ratio
12. Line 118: Thinking about these hypotheses and predictions, I find myself conflicted. Up to this point I haven’t been convinced that mothers can perceive the presence of alien vs resident conspecific (line 112). How is this information transmitted (e.g., through silk, touching with first pair of legs, etc.)? Moreover, there should be more attention dedicated towards building these predictions. My biggest question thus far is why should we expect inter-population interactions to influence maternal effects, specifically alter son ART? The authors describe the complexity of the cost/benefits of interpopulation interactions, many of which are influenced by factors not tested here. Therefore, it is not clear why females should particularly respond in one way or another to a different population, or the direction it responds. Furthermore, it is left unclear as to how adjusting son ARTs should be in response to interpopulation interactions. For example, if the presence of an alien conspecific reduces host-plant defenses, what changes are expected to occur in female reproductive behavior/son ART?
13. Line 181: This had been previously stated in line 88
14. Line 130: Seeing now that the female perception of alien vs resident environment/mate is an assumption, it would be helpful to motivate this assumption here by referencing work showing that there is some level of discrimination in this species (e.g., Clotuche et al 2012)
15. Line 205: suggest starting a separate section here for assessment of son ARTs (the paragraph is very long)
16. Line 242: For predicting the number of offspring produced, why assume a normal distribution and not Poisson or binomial for count data?
17. Line 306: It is not clear if/how these responses are adaptive. There were changes in response to mate and socio-environment, however there is no clear link as to how adaptive these alterations are.
18. Line 318-331: I believe the logic expressed in this paragraph should be incorporated into the introduction which would alleviate a lot of my concerns (comment 12). If the assertion is that maternal effects will be expressed as an adaptive response to the risk/reward of inter-population interactions, please explain how you interpret risk/reward and the responses to each. That is, although you do not have a priori predictions as for which population of females will be at risk, you do expect those females that are at risk to be the population that produce less eggs in the new socio-environment than the resident. Given that, what alterations are expected in response to that risk/reward environment? For example, is there one reproductive tactic to be expressed over another in a given environment (lines 409-417)? This could follow right after suggestion number 11 and, in my opinion, would make this manuscript very streamlined with clear underlying predictions. I think this is a very interesting study but currently it is difficult to follow some of the logic.
19. Line 329: I am curious what you would expect if males from different populations were to compete for access to females. Here you show that Y- males appear to fertilize more eggs, but would they be able to compete directly with G- males.

Reviewer: 2

Comments to the Author(s)

In this paper, the authors demonstrated how maternal social-environment and mate affects their own egg production and sons’ pre-mating behaviour. The full $2 \times 2 \times 2$ factorial design is good to test their hypothesis. However, I cannot understand a couple of experimental designs. Why did the authors expose mothers to their own or alien conspecifics twice (i.e. during juvenile

development and during oviposition)? Why not only during oviposition? And why were mothers exposed to 20 males during juvenile development (line 187)? This is an unusual situation for this mite. I think it would be more convincing if the authors expose mothers to their own or alien conspecifics only during oviposition. The current experimental design is a bit tricky.

In addition, Figure 1 does not give proper information to readers. Can the authors show the treatments with the time course? Also, can they show maternal origin and treatments separately? Exactly saying, the first letter, G and Y, of Y-Y-Y, Y-G-Y, Y-Y-G...etc. are not 'maternal treatments', but 'maternal origin'. Y-Y-Y, Y-G-Y, Y-Y-G...etc. look like there were three treatments.

The colours in Figures are also confusing. In Figures 2-4 & 7, blue colour is used for Y maternal origin and red colour is for G maternal origin, but it is not clear in Figure 1. And if the authors use blue and red colours for G and Y, it is better to use different colours for Figure 5.

Author's Response to Decision Letter for (RSOS-191201.R0)

See Appendix A.

RSOS-191201.R1 (Revision)

Review form: Reviewer 1

Is the manuscript scientifically sound in its present form?

Yes

Are the interpretations and conclusions justified by the results?

Yes

Is the language acceptable?

Yes

Do you have any ethical concerns with this paper?

No

Have you any concerns about statistical analyses in this paper?

No

Recommendation?

Accept as is

Comments to the Author(s)

I read the revised manuscript entitled Spider mite mothers adjust reproduction and sons' alternative reproductive tactics to alien conspecifics. Using a full-factorial experimental design, the authors examined the influence of maternal socio-environment and mate on reproductive output and son mating behavior and reproductive tactics among two populations of spider mites.

I found the revisions undertaken by the authors to be compelling and satisfactory. Particularly, the introduction is clear and succinctly relays the expectations and rationale for the study and

experimental design. The results and interpretation of the study are indeed intriguing and warrant publication.

Review form: Reviewer 2

Is the manuscript scientifically sound in its present form?

Yes

Are the interpretations and conclusions justified by the results?

Yes

Is the language acceptable?

Yes

Do you have any ethical concerns with this paper?

No

Have you any concerns about statistical analyses in this paper?

No

Recommendation?

Accept as is

Comments to the Author(s)

The manuscript has been improved. I have no more comments.

Decision letter (RSOS-191201.R1)

04-Nov-2019

Dear Dr Schausberger,

I am pleased to inform you that your manuscript entitled "Spider mite mothers adjust reproduction and sons' alternative reproductive tactics to alien conspecifics" is now accepted for publication in Royal Society Open Science.

Royal Society Open Science operates under a continuous publication model (<http://bit.ly/cpFAQ>). Your article will be published straight into the next open issue and this will be the final version of the paper. As such, it can be cited immediately by other researchers.

As the issue version of your paper will be the only version to be published I would advise you to check your proofs thoroughly as changes cannot be made once the paper is published.

Kind regards,
Andrew Dunn
Senior Publishing Editor
Royal Society Open Science
openscience@royalsociety.org

on behalf of Dr Alexander Ophir (Associate Editor) and Kevin Padian (Subject Editor)
openscience@royalsociety.org

Associate Editor Comments to Author (Dr Alexander Ophir):

Associate Editor: 1

Comments to the Author:

Dear Dr. Schausberger,

Thank you for your effort toward revising your manuscript. The reviewers felt your paper has addressed their critiques and I agree that it is now a strong paper worthy of publication. Congratulations on a very nice study.

Alex Ophir
Associate Editor RSOS.

Reviewer comments to Author:

Reviewer: 2

Comments to the Author(s)

The manuscript has been improved. I have no more comments.

Reviewer: 1

Comments to the Author(s)

I read the revised manuscript entitled Spider mite mothers adjust reproduction and sons' alternative reproductive tactics to alien conspecifics. Using a full-factorial experimental design, the authors examined the influence of maternal socio-environment and mate on reproductive output and son mating behavior and reproductive tactics among two populations of spider mites.

I found the revisions undertaken by the authors to be compelling and satisfactory. Particularly, the introduction is clear and succinctly relays the expectations and rationale for the study and experimental design. The results and interpretation of the study are indeed intriguing and warrant publication.

Appendix A

Our response to the points raised by the two reviewers is underlined.

Reviewer: 1

Comments to the Author(s)

I read with fascination the manuscript entitled Spider mite mothers adjust reproduction and sons' alternative reproductive tactics to alien conspecifics. The authors investigated an important and increasingly popular topic in evolutionary and behavioral ecology: the influences of transgenerational plasticity. Particularly, the authors investigated how female and son reproductive behavior is altered by the socio-environment and mate origin that the female experiences. The authors found compelling evidence that supports the notion that females do in fact adjust their behavior and the behavior of their (haploid) sons in response to the socio-environment and its perceived risk.

I found the underlying premise for the study intriguing, the methodology and statistics sound, and the results exciting. My only complaint is that I found the introduction fairly difficult to follow. First, I note use of terminology such as "alien" and "invasive" does not seem to fit here as the study concerns conspecifics of differing populations. Second, the build-up to the hypotheses and predictions are difficult to follow. Much of the rationale for any given prediction is not explained until the end of the manuscript, in the discussion. This makes it hard to understand how the authors intend to interpret things such as female perception of risk and what expectation (if any) for how females and males should respond to this risk. I would like to see the introduction for this manuscript modified so that a priori hypotheses are made clear.

We amended the introduction to clarify our hypotheses and predictions. Importantly, we now provide a reference, which was not yet available at submission of the first version of this manuscript, describing how mothers can influence the ARTs of their sons to make them more successful or less successful in mating behavior (reference 39; Schausberger and Sato 2019). We now emphasize already in the introduction that the spider mites can discriminate individuals from their own and the alien population.

1. Line 36: I question if it is necessary to include the discussion of the endosymbiont here given that testing for it was standard procedure and didn't appear to influence the results

Endosymbiont status of the G-population did not have any direct classical effect but may have influenced the inter-population interaction in an indirect way, via the shared leaf. We thus think it is important to mention this possible endosymbiont influence also in the abstract.

2. Line 38: The term "invader" here does not seem to capture the ecological relevance of this question. First, I'm not quite sure which colony should be considered the invader in this scenario (is it the experimental mother or the mates?). Second if we are discussing inter-population mixing, then "immigration" is probably a more accurate term.

We now use either immigrating alien or alien or immigrant throughout the manuscript.

3. Line 69: erroneous comma after plasticity.

Comma deleted (line 69)

4. Also as an aside, in lines 74-76 discusses how we often expect maternal effects to occur largely when the environment of the mother matches that of the offspring. However, line 97 states that inter-population interactions in this species are common (and presumably so is exposure to cues from different populations). If this is the case, can you explain why *T. urticae* would express maternal effects if the environments may be unpredictable? One explanation is that your design is providing very predictable information by exposing females to the same environment at two phases (T-phase and when ovipositing). This link should be made clear somewhere in the MS.

Inter-population interactions are common when considering longer time periods (years) but not so common within shorter time periods that mothers cannot predict the environment their offspring will encounter. Inter-population interactions typically start with arrival of one or a few females in the domain of a resident population.

5. Line 86: citation typo after “phenotypes”

Corrected (line 86)

6. Line 93: This bit of the sentence is confusing. Consider rewording.

Rephrased (lines 90 to 93)

7. Line 100: Are there known behavioral changes when populations interact? Please briefly mention How/if individuals can discern from same vs distinct populations (line 106)? Also due note that your use of the term “invasion” may be better described as “immigration”. Unless the (sub)populations essentially function as colonies, in which case this isn’t made clear. In any case, you use the term “alien” and “invader” interchangeably throughout the MS. I recommend sticking with one (“alien” is in the title and probably more accurately describes these interactions although probably not more so than “immigrant”).

We now mention already in the Introduction that *T. urticae* are known to possess the ability to discriminate individuals from their own and another population (lines 104 to 106). Invading changed to alien or immigrating alien or immigrating throughout the manuscript.

8. Line 103: But this isn’t at all the scenario being tested. For example, part of your experimental design is more like an immigrant female ovipositing on the host plant of a different population.

Immigration by spider mites from one population into another occurs primarily via the arrival of one or a few ovipositing females, because those females are the main dispersers. Thus, the description of this scenario does apply to our experiment.

9. Line 104-107: Please consider breaking this sentence up or rephrasing. It is currently a bit difficult to read. Suggestion: “Resident females may perceive the arrival of alien conspecifics as a threat or an advantage depending on a number of factors. These include...”.

Rephrased as suggested (lines 106 to 109).

10. Line 112: What is meant by females “adjust[ing] their own reproductive behavior”? My understanding up to this point is that females are essentially immobile at the time of copulation. Do females have control over reproductive output? Please clarify.

Rephrased (reproduction instead of reproductive behavior) (line 114). Females have control over the reproductive output, they can decide to let their eggs fertilize by the sperm of their mate or not and adjust the number and sex ratio of the offspring.

11. Line 117: What is meant by 'performance' by ARTs? Do you mean male mating success or the overall expression of different ARTs? And what is meant by "enhanced"? Because there were no clear differences in ART ratio

Clarified and reworded (lines 116 to 123). We now provide a recently published reference (reference 39; Schausberger and Sato 2019), which could not yet be included in the original submission, describing how mothers can influence the ARTs of their sons to make them more or less successful in mating behavior.

12. Line 118: Thinking about these hypotheses and predictions, I find myself conflicted. Up to this point I haven't been convinced that mothers can perceive the presence of alien vs resident conspecific (line 112). How is this information transmitted (e.g., through silk, touching with first pair of legs, etc.)? Moreover, there should be more attention dedicated towards building these predictions. My biggest question thus far is why should we expect inter-population interactions to influence maternal effects, specifically alter son ART? The authors describe the complexity of the cost/benefits of interpopulation interactions, many of which are influenced by factors not tested here. Therefore, it is not clear why females should particularly respond in one way or another to a different population, or the direction it responds. Furthermore, it is left unclear as to how adjusting son ARTs should be in response to interpopulation interactions. For example, if the presence of an alien conspecific reduces host-plant defenses, what changes are expected to occur in female reproductive behavior/son ART?

We now mention, and provide the relevant references early on, that *T. urticae* has kin recognition abilities allowing discriminating individuals from their own and an alien population (lines 104 to 106). Maternal effects are adaptive if mothers can reliably predict the environment their offspring are likely to encounter and prepare them accordingly. Since it pays to respond to immigrants as early and quickly as possible, it should already be the mothers who induce changes in the offspring.

13. Line 181: This had been previously stated in line 88

Changed (line 183)

14. Line 130: Seeing now that the female perception of alien vs resident environment/mate is an assumption, it would be helpful to motivate this assumption here by referencing work showing that there is some level of discrimination in this species (e.g., Clotuche et al 2012)

Done (lines 104 to 106)

15. Line 205: suggest starting a separate section here for assessment of son ARTs (the paragraph is very long)

Done. The whole paragraph has been broken up in several parts (lines 175 to 223).

16. Line 242: For predicting the number of offspring produced, why assume a normal distribution and not Poisson or binomial for count data?

We checked the data distribution before analysis. They follow a normal distribution but not a Poisson or negative binomial distribution.

17. Line 306: It is not clear if/how these responses are adaptive. There were changes in response to mate and socio-environment, however there is no clear link as to how adaptive these alterations are.

In the discussion we provide explanations why these changes are adaptive. For example, since G-individuals create a more favorable socio-environment than Y-individuals, it is adaptive for Y-mothers to produce more offspring in G-environments. Similarly, it is more favorable for G-males to increase their guarding propensity when perceiving unfavorable Y environments, to warrant their influence on offspring production of their G-mates.

18. Line 318-331: I believe the logic expressed in this paragraph should be incorporated into the introduction which would alleviate a lot of my concerns (comment 12). If the assertion is that maternal effects will be expressed as an adaptive response to the risk/reward of inter-population interactions, please explain how you interpret risk/reward and the responses to each. That is, although you do not have a priori predictions as for which population of females will be at risk, you do expect those females that are at risk to be the population that produce less eggs in the new socio-environment than the resident. Given that, what alterations are expected in response to that risk/reward environment? For example, is there one reproductive tactic to be expressed over another in a given environment (lines 409-417)? This could follow right after suggestion number 11 and, in my opinion, would make this manuscript very streamlined with clear underlying predictions. I think this is a very interesting study but currently it is difficult to follow some of the logic.

Judgment of whether immigrating alien conspecifics are considered a threat or advantage to resident conspecifics in the two populations used in our experiments is based on the results. We do not want to retrofit the introduction as if we would have known beforehand which population creates a more favorable environment than the other one. Possible risks and rewards are explained in the introduction in lines 97 to 123. Most importantly, we now provide a reference that describes possible changes in ARTs making the males more or less successful in mating behavior (lines 116 to 123).

19. Line 329: I am curious what you would expect if males from different populations were to compete for access to females. Here you show that Y- males appear to fertilize more eggs, but would they be able to compete directly with G- males.

We assume that Y-males are superior competitors to G-males because they are more eager to guard.

Reviewer: 2

Comments to the Author(s)

In this paper, the authors demonstrated how maternal social-environment and mate affects their own egg production and sons' pre-mating behaviour. The full 2 x 2 x 2 factorial design is good to test their hypothesis. However, I cannot understand a couple of experimental designs. Why did the authors expose mothers to their own or alien conspecifics twice (i.e. during juvenile development and during oviposition)? Why not only during oviposition? And why were mothers exposed to 20 males during juvenile development (line 187)? This is an unusual situation for this mite. I think it would be more convincing if the authors expose mothers to their own or alien conspecifics only during oviposition. The current experimental design is a bit tricky.

We exposed mothers during development AND oviposition to either own or alien conspecifics to increase the exposure time and hence the chance that they perceive and respond to those socio-environments. It is unknown whether maternal effects in spider mites are induced early in life and/or during oviposition but in many animals it is especially the early life experiences that induce long-lasting, including maternal, effects. We exposed the experimental mothers during their juvenile phase to a socio-environment created by adult males to be able discriminating the individuals creating the socio-environment and those growing up in the environment. If we would have used adult females to create the socio-environment, those females would have produced eggs confounding the experimental design; if we would have used juveniles to create the socio-environment, we would not have been able to discriminate juveniles creating the socio-environment from juveniles to become experimental mothers. Since the males were removed before the experimental mothers reached adulthood, there was no sex-related interaction between the juveniles to become experimental mothers and the males creating the socio-environment.

In addition, Figure 1 does not give proper information to readers. Can the authors show the treatments with the time course? Also, can they show maternal origin and treatments separately? Exactly saying, the first letter, G and Y, of Y-Y-Y, Y-G-Y, Y-Y-G...etc. are not 'maternal treatments', but 'maternal origin'. Y-Y-Y, Y-G-Y, Y-Y-G...etc. look like there were three treatments.

We amended figure 1 by inserting lower case letters inside text boxes to better make clear to which property of the eight treatments and to which time phase the boxes refer to. We think that the three letter acronyms are properly representing the nature of the eight treatments. Maternal origin is the first of three properties defining a given treatment.

The colours in Figures are also confusing. In Figures 2-4 & 7, blue colour is used for Y maternal origin and red colour is for G maternal origin, but it is not clear in Figure 1. And if the authors use blue and red colours for G and Y, it is better to use different colours for Figure 5.

We removed the colors in figures 1 and 5 to avoid confusing the use of blue and red for Y and G origins as done in figures 2, 3, 4 and 7.